# Sarcopenia, Obesity, and Sarcopenic Obesity: Relationship with Skeletal Muscle Phenotypes and Single Nucleotide Polymorphisms

**DOI:** 10.3390/jcm10214933

**Published:** 2021-10-25

**Authors:** Praval Khanal, Alun G. Williams, Lingxiao He, Georgina K. Stebbings, Gladys L. Onambele-Pearson, Martine Thomis, Hans Degens, Christopher I. Morse

**Affiliations:** 1Musculoskeletal Science and Sports Medicine Research Centre, Department of Sport and Exercise Sciences, Manchester Metropolitan University Institute of Sport, Manchester Metropolitan University, Manchester M15 6BH, UK; a.g.williams@mmu.ac.uk (A.G.W.); lingxiao.he@hotmail.com (L.H.); G.Stebbings@mmu.ac.uk (G.K.S.); G.Pearson@mmu.ac.uk (G.L.O.-P.); c.morse@mmu.ac.uk (C.I.M.); 2Physical Activity, Sports & Health Research Group, Department of Movement Sciences, KU Leuven, 3001 Leuven, Belgium; martine.thomis@kuleuven.be; 3Institute of Sport, Exercise and Health, University College London, London W1T 7HA, UK; 4Musculoskeletal Science and Sports Medicine Research Centre, Department of Life Sciences, Manchester Metropolitan University Institute of Sport, Manchester Metropolitan University, Manchester M15 6BH, UK; h.degens@mmu.ac.uk; 5Institute of Sport Science and Innovations, Lithuanian Sports University, LT-44221 Kaunas, Lithuania

**Keywords:** elderly, sarcopenic obese, neuromuscular, single-nucleotide polymorphisms

## Abstract

Obesity may aggravate the effects of sarcopenia on skeletal muscle structure and function in the elderly, but no study has attempted to identify the gene variants associated with sarcopenia in obese women. Therefore, the aims of the present study were to: (1) describe neuromuscular function in sarcopenic and non-sarcopenic women with or without obesity; (2) identify gene variants associated with sarcopenia in older obese women. In 307 Caucasian women (71 ± 6 years, 66.3 ± 11.3 kg), skeletal muscle mass was estimated using bioelectric impedance, and function was tested with a 30 s one-leg standing-balance test. Biceps brachii thickness and vastus lateralis cross-sectional area (VL_ACSA_) were measured with B-mode ultrasonography. Handgrip strength, maximum voluntary contraction elbow flexion (MVC_EF_), and knee extension torque (MVC_KE_) were measured by dynamometry, and MVC_KE_/VL_ACSA_ was calculated. Genotyping was performed for 24 single-nucleotide polymorphisms (SNPs), selected based on their previous associations with muscle-related phenotypes. Based on sarcopenia and obesity thresholds, groups were classified as sarcopenic obese, non-sarcopenic obese, sarcopenic non-obese, or non-sarcopenic non-obese. A two-way analysis of covariance was used to assess the main effects of sarcopenia and obesity on muscle-related phenotypes and binary logistic regression was performed for each SNP to investigate associations with sarcopenia in obesity. There were no significant obesity * sarcopenic status interactions for any of the investigated muscle-related phenotypic parameters. Neither sarcopenia nor obesity had a significant effect on biceps brachii thickness, but sarcopenia was associated with lower VL_ACSA_ (*p* = 0.003). Obesity was associated with lower MVC_EF_ (*p* = 0.032), MVC_KE_ (*p* = 0.047), and MVC_KE_/VL_ACSA_ (*p* = 0.012) with no significant effect of sarcopenia. Adjusted for age and height, three SNPs (*ACTN3* rs1815739, *MTHFR* rs1801131, and *MTHFR* rs1537516) were associated with sarcopenia in obese participants. Sarcopenia was associated with a smaller muscle size, while obesity resulted in a lower muscle quality irrespective of sarcopenia. Three gene variants (*ACTN3* rs1815739, *MTHFR* rs1801131, and *MTHFR* rs1537516) suspected to affect muscle function, homocysteine metabolism, or DNA methylation, respectively, were associated with sarcopenia in obese elderly women. Understanding the skeletal muscle features affected by sarcopenia and obesity, and identification of genes related to sarcopenia in obese women, may facilitate early detection of individuals at particular risk of sarcopenic obesity.

## 1. Introduction

Ageing is associated with an increase in visceral and intramuscular fat [1] and a decrease in skeletal muscle mass [2]. The co-existence of low muscle mass and higher fat is termed sarcopenic obesity (SO) [3], a condition that is most prevalent in the older populations [4], particularly in women [5]. Sarcopenia and obesity are interlinked and share some common pathophysiology. For instance, both are related to hormonal alteration, low physical activity, elevated oxidative stress and pro-inflammatory cytokines, and insulin resistance [6]. It has been suggested that the synergistic effect of sarcopenia and obesity in SO exacerbates the condition compared to sarcopenia or obesity alone [7,8]. This has been evidenced through the higher all-cause mortality in SO than sarcopenia or obesity alone in men [9] and the aggravated detrimental effect on physical performance [10].

The prevalence of SO can range from 3.6 to 94.0% in the same population of elderly women depending on the applied sarcopenia and obesity thresholds/indices [11], an issue that plagues many study definitions [12]. Lower handgrip strength (HGS) is an initial confirming diagnosis of sarcopenia [13]. SO studies that define sarcopenia through lower HGS and muscle mass show a better prediction of physical capacity/functional status in an elderly population [14,15,16,17], compared to studies using a lower muscle mass threshold alone [18,19,20,21]. Similarly, in the context of obesity identification, the threshold of percentage body fat (BF%) may be more valid than body mass index (BMI) as BMI may not change noticeably in some cases, despite a significant change in muscle and fat mass with ageing [22]. Therefore, the inclusion of thresholds for lower muscle strength together with lower muscle mass to detect sarcopenia, and use of percentage of adiposity in obesity classifications, may best distinguish between non-sarcopenic non-obese, sarcopenic, obese, and SO older people [23,24]. Despite the inconsistency in SO definitions, SO has been associated with limitations to perform activities of daily living [3], metabolic disorders [25], cardiovascular diseases (CVDs) [9], physical performance [26] impaired pulmonary function [27], and increased risk of all-cause mortality [28].

In the elderly, impairments in neuromuscular function are likely dependent on the severity of muscle wasting and obesity. For instance, the impact of loss of muscle mass on muscle function is well established [29,30], and similarly obesity is known to impact neuromuscular function negatively in the elderly [31,32]. The fact that some elderly people are SO while others are not suggests that SO is the consequence of the interplay between factors that influence both muscle and obesity related phenotypes, such as physical activity, diet, sedentary behavior, and genetics [33,34,35]. Several studies have associated single-nucleotide polymorphisms (SNPs) with muscle mass and strength in the elderly [36,37], and recently our group found that sarcopenia in elderly women was associated with polymorphisms in *FTO*, *TRHR*, *ESR1*, and *NOS3* [38]. Similarly, adiposity and obesity measures have been associated to SNPs such as *FTO* and *PPARG2* in the elderly [39]. Yet, the interrelation of SNPs and SO has not been investigated. Since sarcopenia and obesity each have been associated with adverse outcome measures in the elderly [40,41], an assessment of the modulating role of gene variants should include sarcopenia and obesity groups alongside an SO group. The aim of the present study was therefore to: (1) describe neuromuscular function in sarcopenic and non-sarcopenic women with or without obesity, and (2) identify gene variants associated with sarcopenia in obese elderly women.

## 2. Methodology

### 2.1. Participants

Elderly women (*n* = 307, aged 71 ± 6 years) were recruited from community groups in England, such as the University of Third Age (U3A) and from word-of-mouth contacts. The participants were 60+ years, free from any issues that affected their daily activities and physical independence, and had no known neuromuscular or cardiovascular disorders. Informed consent was collected from participants. Study protocols for the project followed the Declaration of Helsinki guidelines and were approved by the ethics committee of Manchester Metropolitan University.

### 2.2. Procedures

Procedures for body composition, muscle size, functional tests, and genotyping are described in detail elsewhere [38,42], with a brief overview provided below.

### 2.3. Body Composition and Muscle Size

Bio-impedance analysis (Bodystat 1500MDD, Isle of Man, UK) was used to estimate the skeletal muscle mass (SMM) and BF% of the participants [38]. Adhesive electrodes were connected to the dorsum of the right hand and leg with participants in a supine position. SMM was estimated with a previously validated equation [43] and BF% was recorded. Skeletal muscle index (SMI) was calculated as SMM/height^2^ where height is in m.

Biceps brachii thickness was measured by ultrasound (MyLab™Twice, Esaote Biomedical, Genoa, Italy). In short, the participants were asked to relax and hang their dominant arm at their side and then the sagittal plane ultrasound scan was performed at 60% length from the proximal end of the humerus [44]. Mean muscle thickness was measured with digitizing software (ImageJ 1.45, National Institute of Health, Bethesda, MD, USA) at three sites [45] (proximal, middle, and distal ends of the captured image) and recorded as biceps brachii thickness.

To assess the anatomical cross-sectional area of the vastus lateralis muscle (VL_ACSA_), the same ultrasound was used to scan in a transverse plane at 50% of VL length. During the procedure, the ultrasound probe was moved gently over the echo-absorptive markers (placed from medial to lateral border of muscle) with minimal pressure applied, and the scan was recorded as a video file. The recorded video was split into different individual images, and the entire VL_ACSA_ was reconstructed by combining the individual images between the contiguous intervals between each shadow cast by markers. For the measurement, digitizing software (ImageJ 1.45, National Institute of Health, Bethesda, MD, USA) was used to draw around the visible aponeurosis of the reconstructed VL to estimate ACSA. The reliability and validity of this method were previously reported as high (ICC > 0.99) when compared with MRI [46].

### 2.4. Muscle Strength and Quality Measurement

Handgrip strength (HGS) was measured for both the right and left hands with a dynamometer (JAMAR plus, JLW Instruments, Chicago, IL, USA). Three trials were performed with each hand and the highest value from either hand was recorded [38]. The ICC of HGS with this procedure is 0.99 [47].

For the measurement of maximum voluntary contraction elbow flexion torque (MVC_EF_) and maximum voluntary contraction knee extension torque (MVC_KE_), a customized dynamometer with a load cell (Zemic, Eten-Leur, the Netherlands) was used. Both procedures were performed on the participant’s dominant side, identified by self-report. MVC_EF_ was performed at 60° flexion (0° is a straight position). Three trials were performed and the maximum of the three attempts was recorded and converted to MVC_EF_ torque as:MVC_EF_ = Force × Radius length × Cos30°

Similarly, three trials were used to assess MVC_KE_ at 120° knee extension (180° is a full knee extension) with MVC_KE_ torque calculated from the highest value as:MVC_KE_ = Force × (distance from rotation point of dynamometer to ankle strap) × Cos30°

Subsequently, lower-limb muscle quality was defined as the knee strength relative to VL_ACSA_ (MVC_KE_/VL_ACSA_).

### 2.5. One-Leg Standing-Balance Test

As a measure of balance and for the purposes of classifying participants (see statistics part below), the one-leg standing-balance test (OLST) was performed barefoot on the right leg regardless of dominance, consistent with previous studies [48,49]. Participants were instructed to stand straight, with their left knee flexed at ~90°, for up to a maximum of 30 s. Participants completed up to three attempts if they failed to achieve 30 s, and the maximum time achieved was recorded as their OLST score [50].

### 2.6. Selection of Single-nucleotide Polymorphisms, Sample Collection, DNA Extraction and Genotyping

A total of 24 SNPs were selected for the present study (Appendix A) based on a number of reasons, including the presence of extant literature associating those SNPs with similar phenotypes (even if in different populations), and the likelihood of affecting relevant muscle phenotypes via transcriptional differences as reported in previous studies. Some SNPs were also included due to conflicting results regarding associations with relevant phenotypes.

Blood or saliva samples were collected for genetic analysis. A QIAcube robot, QIAamp DNA Blood Mini kit, and standard spin column protocol (Qiagen, Crawley, UK) were used to isolate DNA from the collected samples. A Fluidigm EP1 system (Fluidigm, Cambridge, UK) was used for initial genotyping of the 24 SNPs. When there was an error (such as no agreement between duplicates), a StepOnePlus (Applied Biosystems^®^, Paisley, UK) was later used. The PCR conditions using the EP1 followed the manufacturer’s instructions. In short, the thermocycling protocol was an initial 120 s at 95 °C followed by 45 cycles of denaturation for 2 s at 95 °C and then annealing and extension for 20 s at 60 °C. Genotypes were identified based on end-point fluorescence, whereby TaqMan assays (Applied Biosystems, Paisley, UK) included VIC^®^ and FAM^®^ dyes for all SNPs. Similarly, for the StepOnePlus, the thermocycling conditions were an initial 20 s at 95 °C followed by 50 cycles of denaturation for 3 s at 95 °C, then annealing and extension for 20 s at 60 °C. Genotyping and interpretation were based on intensity of VIC^®^ and FAM^®^ intensity and visualization was performed using cluster plots.

## 3. Statistics

SPSS 26.0 (SPSS, Chicago, IL, USA) was used for all analyses. Participants were characterized into four non-overlapping groups based on thresholds for sarcopenia and obesity. Sarcopenia was defined as an SMI < 6.76 kg/m^2^ [51] and lower HGS as suggested by EWGSOP [52]. Receiver operating curve (ROC) analysis was performed to determine the sarcopenic threshold for lower HGS based on an OLST balance of <5 s [53]. This resulted in a good classification (AUC = 0.80) of lower HGS with OLST and identified the HGS threshold of 28.35 kg (~28.5 kg) (sensitivity 65%, specificity 82%) as a measure of low muscle strength. Obesity was defined as BF% > 38% [28].

Based on the above thresholds for sarcopenia and obesity, SO individuals were defined as those with SMI < 6.76 kg/m^2^, HGS < 28.5 kg and BF% > 38%. Non-sarcopenic obese were defined as individuals who were non-sarcopenic (either SMI ≥ 6.76 kg/m^2^ or HGS > 28.5 kg or both) but had BF% > 38%. Sarcopenic non-obese were defined as individuals who were sarcopenic (SMI < 6.76 kg/m^2^ and HGS < 28.5 kg) and had BF% ≤ 38%. The non-sarcopenic non-obese group was defined as individuals who were non-sarcopenic (either SMI ≥ 6.76 kg/m^2^ or HGS > 28.5 kg or both) and had BF% ≤ 38%. The classification of participants into four groups is shown in Figure 1 and adopted an approach used previously [5,18,54].

A Shapiro–Wilk test was performed to test whether the data were normally distributed. Variables that were not normally distributed (biceps brachii thickness, MVC_EF_, and MVC_KE_/VL_ACSA_) were log-transformed before further statistical analysis. To identify covariates, Pearson correlation coefficients were calculated between age and height, and body composition indices (BF% and SMI). A two-way analysis of covariance (ANCOVA) was used to assess the main effects of sarcopenia and obesity on muscle-related phenotypic outcome measures, where a sarcopenia * obesity interaction would indicate that the effects of sarcopenia differed between the obese and non-obese, or that the effect of obesity was different in sarcopenic and non-sarcopenic women.

Binary logistic regression was performed for each individual SNP using the covariates to investigate their association with sarcopenia in the obese elderly. Where *p* < 0.15, the rarer homozygotes were combined with heterozygotes (in both dominant and recessive models) and two-group analysis was performed. Similarly, when very few participants were in one of the homozygous groups, the group was combined with the heterozygotes [55]. Odds ratio (OR) was used to estimate effect size for significant SNP-groups associations. In all analyses, *p* < 0.05 was considered significant. Only significant SNP associations are reported.

## 4. Results

### 4.1. General Characteristics, Prevalence, and Differences in Muscle Size, Strength, and Quality among Groups

The prevalence of sarcopenic-obese, non-sarcopenic obese, sarcopenic non-obese, and non-sarcopenic non-obese groups was 25.1%, 57.3%, 2.3%, and 15.3%, respectively, in the present elderly women. There were significant differences in age, height, body mass, BMI, SMI, HGS, OLST, and BF% between the groups (Table 1).

There were significant correlations between BF% and age (r = 0.290, *p* <0.001), and BF% and height (r = −0.250, *p* < 0.001). Table 2 shows the data on muscle size and strength. There was no significant obesity * sarcopenic status interaction for any of the investigated muscle-related phenotypic parameters, indicating that the effect of sarcopenia did not differ according to obesity status, and that the effect of obesity did not differ according to sarcopenia status. There was no significant impact of obesity and sarcopenia on biceps brachii thickness (*p* > 0.05). Sarcopenia was associated with lower VL_ACSA_ (*p* = 0.003), but there was no significant effect of obesity (*p* = 0.074). Obesity was associated with a lower MVC_EF_ (*p* = 0.032), MVC_KE_ (*p* = 0.047), and MVC_KE_/VL_ACSA_ (*p* = 0.012) with no significant effect of sarcopenia on these parameters (*p* > 0.05).

### 4.2. Associations of Gene Variants with Sarcopenia in Obese Elderly Women

The genotyping success rate was > 99% in the obese subgroups (*n* = 77 + 176). All genotype distributions were in Hardy–Weinberg equilibrium (*p* > 0.05). Among the 24 SNPs, three were associated with sarcopenia: *ACTN3* rs1815739, *MTHFR* rs1801131, and *MTHFR* rs1537516 (Table 3) in obese elderly. The genotype distribution of these three SNPs is shown in Appendix A.

Binary logistic regression adjusted for age and height showed that *ACTN3* rs1815739 CC homozygotes had 1.8 times the odds of being in the sarcopenic group than T-allele carriers (OR = 1.84, 95% confidence interval [CI] = 1.04–3.28, *p* = 0.037) in an obese population. *MTHFR* rs1801131 G-allele carriers had 1.9 times the odds of being sarcopenic than TT homozygotes (OR = 1.85, 95% CI = 1.05–3.27, *p* = 0.034) among obese elderly. Similarly, *MTHFR* rs1537516 A-allele carriers had 2.8 times the odds of being sarcopenic than GG homozygotes (OR = 2.75, 95% CI 1.41–5.36, *p* = 0.003, Table 3) in obese elderly women.

## 5. Discussion

The present study showed that sarcopenia was associated with lower-limb muscle atrophy, while obesity was associated with reduced upper/lower-limb muscle strengths. Obesity, but not sarcopenia, was associated with lower muscle quality. We also identified that within the 253 obese elderly women, there were three SNPs that were associated with sarcopenia, namely, *ACTN3* rs1815739, *MTHFR* rs1801131, and *MTHFR* rs1537516. Expanding the knowledge of the impact of polymorphisms on skeletal muscle phenotypes during sarcopenia and obesity in elderly women may help in early identification of individuals at particular risk of sarcopenic obesity.

The 25% prevalence of SO observed in the present elderly population is consistent with other studies reporting SO prevalence across different populations [27,28,56,57,58]. However, it should be noted that, despite similar SO prevalence, studies have used different indices of sarcopenia and obesity classification, which can result in a prevalence of SO in a given population being 4% or even as high as 94% depending on the sarcopenia and obesity definitions/thresholds used [11]. The present study was the first to investigate the association of muscle size, strength, and quality measures in different groups, ranging from non-sarcopenic non-obese to sarcopenic obese. Although we did not observe any effect of sarcopenia and obesity on biceps brachii thickness in the present elderly women, sarcopenia was associated with lower VL_ACSA_. This association of sarcopenia with lower VL_ACSA_ reflects the muscle atrophy in terms of mass/size in sarcopenic individuals [59,60]. We observed in elderly women that obesity is associated with a reduction in upper-limb (MVC_EF_) and lower-limb (MVC_KE_) muscle strength. This is somewhat surprising as most studies report a higher muscle strength in obese [32,61]. We have no explanation for this unexpected finding, but we did see a lower MVC_KE_/VL_ACSA_ reflecting a lower muscle quality that has regularly been reported in obesity [62,63] and has been suggested to be due to accumulation of intramuscular fat, impaired muscle activation [32], and systemic inflammation [32,64]. Our findings suggest that obesity causes a reduced muscle quality, even in non-sarcopenic older women. The reduction in muscle function has been previously associated with an increased risk of falls [65] and disability in activities of daily living [66].

Few studies have attempted to identify the genes associated with sarcopenia [38,67,68,69]. It should be noted that the SNPs showing associations in the present study (*ACTN3* rs1815739, *MTHFR* rs1801131, and *MTHFR* rs1537516) are different from our previous study that identified associations of *FTO, TRHR, ESR1*, and *NOS3* gene variants with sarcopenia [38]. A possible explanation is that the previous study used a different definition of sarcopenia [38]. The present study identified elderly women with obesity who were *ACTN3* rs1815739 RR homozygotes to be at a nearly-2-fold higher risk of sarcopenia than X-allele carriers. This association is consistent with a previous study that observed female > 75-year-old R-allele carriers had a 2.8-fold higher risk of sarcopenia [69], but differs from another study that reported XX as the risk genotype for sarcopenia in older men and women [70]. *ACTN3* rs1815739 has been widely studied with skeletal muscle phenotypes in different populations, with varying results [71]. Although the *ACTN3* R allele mostly favors better athletic (sprint) performance [72] and higher muscle mass and strength phenotypes [73], there are also studies that have reported the X allele as beneficial for fat-free mass in older people or peak power in non-athletic populations [74,75]. The putative effect of *ACTN3* genotype has also shown different results at baseline and post training. For instance, XX homozygotes had greater knee extensor peak strength while the gain after the training was greater for RR homozygotes in women [75]. Despite this *ACTN3* rs1815739 X where the arginine (position 577) is replaced by a premature stop codon, there is no evidence that the deficit of alpha-actinin directly causes muscle-related disease [76], although recent evidence suggests a potential modulating role [77]. Furthermore, *ACTN3* X as a beneficial allele is also consistent with a previous association with longevity [78] and improved oxidative metabolism in *ACTN3* knockout mice [79]. Thus, the relationships of *ACTN3* rs1815739 with muscle performance and health-related phenotypes are complex and seem dependent on context including population type.

The current study was the first to find an association of *MTHFR* gene variants (rs1537516 and rs1801131) with risk factors for sarcopenia in obese elderly women. *MTHFR* rs1537516 A-allele carriers had a >2-fold higher risk than GG homozygotes. As *MTHFR* plays an important role in methylation and thus in epigenetic control [80], the association of rs1537516 with sarcopenia in an obese population may occur via post-transcriptional regulation. However, we urge future studies to investigate the functional relevance of this SNP with sarcopenia. The minor A allele has been previously observed as a risk factor for higher diastolic blood pressure [81]. Similarly, *MTHFR* rs1801131 G-allele carriers had a 1.8-fold higher risk of sarcopenia than TT homozygotes in our elderly female population. Although the GG genotype was advantageous for speed and strength performance in young Russian and Polish athletes [82], our observation may be explained by the association of the GG genotype with higher plasma homocysteine concentration [83]. Previous studies have associated elevated homocysteine with lower physical activity [84] and muscle strength [85]—characteristics observed in sarcopenic populations [52,86].

It is established that the degree and severity of sarcopenia and obesity increases with age [13,87], despite individual differences related to favorable gene variants as well as dietary or exercise habits. Unlike research into athletic performance, where age at the time of sampling does not affect analysis of associations with elite performance while young [88], comparable studies of the elderly may be affected by participant age due to additional associations with longevity and health status. Accordingly, we acknowledge that there was likely to be a selection bias in the current study as we recruited elderly women who were still the healthiest of those within the broader population living independently. However, we suspect that studying elderly women who are more impaired would only stress the extremes of our phenotypes, potentially exposing more and/or stronger genetic associations.

Although the use of an adiposity threshold for the categorization of obesity is commonly used [23,24], the thresholds for sarcopenia are still debatable and responsible for population variance in prevalence [38,89]. In the present study, sarcopenia was defined by HGS and muscle mass cut-offs in line with EWGSOP [52]. We should however acknowledge that our novel approach to establishing a low HGS threshold adopted a functional threshold of OLST < 5 s. In contrast to others who have used a gait speed threshold of 0.8 m/s [90,91], we suggest our approach of using a standing balance threshold is also valid as it was able to make a good prediction for HGS (AUC = 0.80) with sensitivity of 65.0 and specificity of 82.0, when compared with a previous study that showed sensitivity of 76.9 and specificity of 62.5 using gait speed threshold [90].

The current study has some other limitations. Firstly, participants were distributed unevenly between groups, meaning the sarcopenic non-obese group possessed few participants. This uneven group distribution could be overcome with a larger population group. Based on a power analysis for MVC_EF_, we would need at least 21 people in a group to detect a 23.4% difference between groups with an α < 0.05 and a β > 0.80. Our overall sample size would exceed *n* = 900. Secondly, one needs to be cautious to extend our observations to men, as the current study is focused on elderly women, although it is likely that the action of most gene variants will not differ between men and women. Our decision to limit the investigation to a single sex can be justified by the earlier onset of sarcopenia-related problems with daily life in women than men [92]. The same argument applies to the fact that we only studied Caucasians, and findings may not be generalized to people with different geographic ancestry. We contend that this is a limitation of extrapolation of results, rather than a real limitation of method, as the genetic differences between people within a geographic ancestry group are larger than the typical genetic differences between geographic ancestry groups [93]. The current study used BIA for the estimation of skeletal muscle mass, a technique that is reported to overestimate skeletal muscle mass by 0.6 kg in the elderly [94]. Despite this shortcoming we used BIA as it is a cheap, accessible and radiation-free method that has been validated with DXA [95], and adopted on numerous occasions for large population studies of elderly muscle mass [96,97]. Finally, this investigation was limited to the 24 studied SNPs, and in fact it is likely that many other SNPs and other types of genetic variants, which we did not investigate, are also associated with the phenotypes we assessed.

## 6. Conclusions

Lower-limb muscle atrophy is a characteristic of sarcopenia, while obesity in older women was associated with lower upper- and lower-limb muscle strength. Three gene variants (*ACTN3* rs1815739, *MTHFR* rs1801131, and *MTHFR* rs1537516) previously reported to affect muscle function, homocysteine metabolism, or suspected to affect DNA methylation, were associated with sarcopenia in obese elderly women. Understanding the neuromuscular features of sarcopenia and obesity and genes related to sarcopenia in obese elderly women may facilitate early detection of individuals at particular risk of sarcopenic obesity.

## Figures and Tables

**Figure 1 jcm-10-04933-f001:**
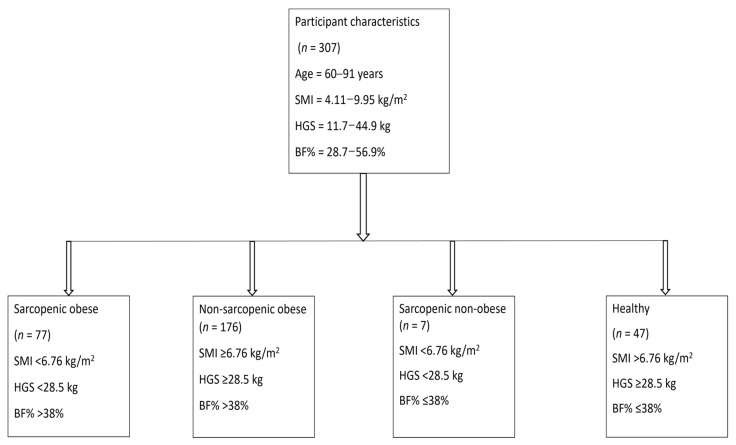
Flowchart showing the classification of participants into four categories: sarcopenic-obese, non-sarcopenic obese, sarcopenic non-obese, and non-sarcopenic non-obese groups. Abbreviations: SMI, skeletal muscle index; HGS, handgrip strength; BF%, body fat percent.

**Table 1 jcm-10-04933-t001:** Physical, sarcopenic, and obesity-related characteristics among groups.

Variables	Overall Characteristics (*n* = 307)	Sarcopenic Obese (*n* = 77)	Non-Sarcopenic Obese (*n* = 176)	Sarcopenic Non-Obese (*n* = 7)	Non-Sarcopenic Non-Obese (*n* = 47)
Age (years)	70.7 (70.1–71.4)	72.5 (71.3–73.8) ^4^	70.8 (70.0–71.7) ^4^	68.7 (65.2–72.2)	67.6 (66.3–69.0) ^1,2^
Height (m)	1.60 (1.59–1.60)	1.58 (1.57–1.59) ^2,4^	1.60 (1.59–1.61) ^1^	1.63 (1.58–1.67)	1.62 (1.61–1.64) ^1^
Body mass (kg)	66.3 (65.01–67.6)	62.1 (60.1–64.0) ^2^	70.5 (68.7–72.2) ^1,3,4^	55.2 (48.6–61.7) ^2^	59.6 (57.5–61.6) ^2^
BMI (kg/m^2^)	25.9 (25.4–26.4)	24.8 (24.1–25.6) ^2,3,4^	27.5 (26.9–28.1) ^1,3,4^	20.8 (19.1–22.5) ^1,2^	22.5 (22.0–23.1) ^1,2^
SMI (kg/m^2^)	6.56 (6.47–6.65)	5.96 (5.85–6.08) ^2,4^	6.71 (6.59–6.83) ^1^	6.21 (5.83–6.60) ^4^	7.00 (6.80–7.20) ^1,3^
HGS (kg)	29.9 (29.3–30.5)	25.3 (24.7–25.9) ^2,4^	31.0 (30.3–31.7) ^1,3,4^	25.8 (23.6–28.0) ^2,4^	34.0 (32.7–35.2) ^1,2,3^
OLST (s)	23 (22–24)	20 (18–23) ^2,4^	23 (22–25) ^1^	22 (14–30)	29 (28–30) ^1^
BF%	42.6 (42.1–43.2)	44.4 (43.5–45.3) ^3,4^	44.1 (43.5–44.7) ^3,4^	36.1 (34.6–37.7) ^1,2^	35.0 (34.5–35.6) ^1,2^

Abbreviations: BMI, body mass index; SMI, skeletal muscle mass index; HGS, handgrip strength; OLST, one-leg standing-balance test; BF, body fat. ^1,2,3,4^ denote differences from sarcopenic-obese, non-sarcopenic obese, sarcopenic non-obese, and non-sarcopenic non-obese groups, respectively. Data are means (95% CI).

**Table 2 jcm-10-04933-t002:** Muscle size, strength, and quality among groups (phenotype data are presented as means (95% confidence intervals).

Variables	Sarcopenic Obese (*n* = 77)	Non-Sarcopenic Obese (*n* = 176)	Sarcopenic Non-Obese (*n* = 7)	Non-Sarcopenic Non-Obese (*n* = 47)	*p*-Value Obesity	*p*-Value Sarcopenia	*p*-Value Sarcopenia * Obesity
Biceps brachii thickness (cm)	1.66 (1.59–1.74)	1.81 (1.76–1.86)	1.78 (1.51–2.05)	1.68 (1.62–1.75)	0.544	0.933	0.173
VL_ACSA_ (cm^2^)	14.7 (14.0–15.5)	17.1 (16.6–17.6)	13.9 (10.5–17.3)	16.4 (15.4–17.4)	0.074	0.003	0.861
MVC_EF_ (N·m)	21.5 (20.3–22.8)	25.4 (24.6–26.1)	25.7 (19.2–32.2)	27.8 (25.1–30.4)	0.032	0.083	0.251
MVC_KE_ (N·m)	48.2 (44.6–51.8)	56.6 (53.9–59.2)	66.1 (44.9–87.3)	62.9 (55.9–70.0)	0.047	0.498	0.252
MVC_KE_/VL_ACSA_ (N·m/cm^2^)	3.39 (3.10–3.68)	3.37 (3.20–3.54)	4.88 (3.34–6.42)	3.89 (3.48–4.30)	0.012	0.077	0.156

Abbreviations: VL_ACSA_, vastus lateralis anatomical cross-sectional area; MVC_EF_, maximum voluntary contraction elbow flexion torque; MVC_KE_, maximum voluntary contraction knee extension torque. Data are means (95% CI). ^1,2,3,4^ indicate difference from sarcopenic obese, non-sarcopenic obese, sarcopenic non-obese and non-sarcopenic non-obese, respectively at *p* ≤ 0.05. *, standard in statistical notation to denote interaction between the grouping variables.

**Table 3 jcm-10-04933-t003:** Association of single-nucleotide polymorphisms with sarcopenia in obese women.

SNPs	B	S.E (β)	Wald’s χ^2^	p	OR	95% CI	Risk Genotypes
*ACTN3* rs1815739	0.612	0.294	4.341	0.037	1.84	1.04–3.28	CC
*MTHFR* rs1801131	0.614	0.291	4.470	0.034	1.85	1.05–3.27	GG + GT
*MTHFR* rs1537516	1.011	0.341	8.811	0.003	2.75	1.41–5.36	AA + AG

## Data Availability

The datasets generated during and/or analyzed during the present study are available from the corresponding author on reasonable request.

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
