# Peer review of "Sarcopenia, Obesity, and Sarcopenic Obesity: Relationship with Skeletal Muscle Phenotypes and Single Nucleotide Polymorphisms"

_jcm, 2021, doi:10.3390/jcm10214933_

Round 1

Reviewer 1 Report

In this paper, Dr. Khanal et al aimed to describe neuromuscular function in Caucasian elderly women (sarcopenic and non-sarcopenic, with or without obesity), recruited from community groups in England, and to identify genetic variants associated with sarcopenia in obese elderly women. The authors considered several measures reflecting body composition and muscle size (skeletal muscle mass and muscle thickness), as well as muscle strength and quality (upper/lower limb muscle strength, and limb muscle quality). They did not identify significant interactions between obesity and sarcopenia on any of these measures. They found that sarcopenia was associated with lower-limb muscle atrophy while obesity was nominally associated with reduced upper and lower-limb muscle strengths and with lower muscle quality. Finally, the authors detected nominal associations of three SNPs in ACTN3 and MTHFR with sarcopenia in obese elderly women, after adjusting for age and sex.
The major concern is about the limited study sample size resulting in small and unbalanced subgroups and the modest associations detected in this study. Important clarifications are needed in the Method section to make sure the assumptions of the statistical methods are not violated. The authors should also better justify how the limited number of SNPs in their genetic analyses was selected.
Below are further comments the authors should consider to improve the manuscript.

Major
-    The total sample size of the study is small and dividing the data in four categories make some groups very small (one group has even less than 10 participants). The authors should provide information on how they verified that all assumptions of ANCOVA or logistic regressions were met before analysis. Have the authors also done some power calculations based on their sample size?
-    The authors did not mention or perform any correction for multiple testing (Bonferroni or FDR). This should be discussed/justified.
-    The authors need to provide more details on how the 24 SNPs were selected (studies listed in Suppl Table 1). What were the criteria chosen by the authors to select these publications? Why the largest GWAS of handgrip, sarcopenia, lean body mass, and muscle weakness were not included?
-    Limitation section of the Discussion: The authors should discuss more how the small sample sizes and the unbalanced distribution of participants across groups may have affected their results. They should also mention that the study was conducted only in Caucasian participants and thus the results may not generalize to other ancestries or ethnicities.
-    Abstract and Conclusion section: “Three gene variants linked to affect muscle function, homocysteine metabolism or DNA methylation respectively”. Are the MTFHR SNPs methylation QTLs or is it the gene itself that has been reported by EWAS?
-    The authors used Pearson correlations to identify relevant covariates for BF and SMI. They only considered two (age and height). Height is already included in the definition of SMI. They could have used more sophisticated methods that can account for the joint effect of multiple covariates at the same time. The authors did not provide the results of the Pearson correlation analysis for sarcopenia in the manuscript.
-    Line 256: “A possible explanation is that the previous study reported used a different definition of sarcopenia”. Could the authors clarify how the two definitions of sarcopenia differ and why they changed their definition of sarcopenia?
-    More information is needed in the abstract about the statistical analyses performed and the selection of SNPs.

Minor
-    Abstract and Results section: “There were no significant obesity * sarcopenic status interaction for any of the parameters” please rephrase/clarify what are these parameters
-    Please replace the following terminology in the paper: “linked” and “interlinked”
-    In the description of the genetic analyses in the Method section, the authors did not mention that the analyses were performed only in obese women. Please add this information.
-    Suggestion to define sarcopenia in the abstract
-    Suggestion to use “describe” instead of “identify” for the first sub-aim in the abstract, as was done at the end of the introduction
-    Lines 175-176: “A two-way Analysis of covariance (ANCOVA) was used to assess the main effects of sarcopenia and obesity“. The outcomes are missing (effect of X on Y). Please add this information.
-    Line 201 “indicating that all observed effects of sarcopenia are the same” please rephrase
-    Line 248: “Our findings on muscle function and muscle quality suggest that sarcopenia and obesity act synergistically” please use another term as no interaction was significant
-    Table 2: please clarify what the numbers in each cell (column 1-4) represent (mean (95% CI)), and maybe also consider revising the title.

Author Response

Reviewer 1:

In this paper, Dr. Khanal et al aimed to describe neuromuscular function in Caucasian elderly women (sarcopenic and non-sarcopenic, with or without obesity), recruited from community groups in England, and to identify genetic variants associated with sarcopenia in obese elderly women. The authors considered several measures reflecting body composition and muscle size (skeletal muscle mass and muscle thickness), as well as muscle strength and quality (upper/lower limb muscle strength, and limb muscle quality). They did not identify significant interactions between obesity and sarcopenia on any of these measures. They found that sarcopenia was associated with lower-limb muscle atrophy while obesity was nominally associated with reduced upper and lower-limb muscle strengths and with lower muscle quality. Finally, the authors detected nominal associations of three SNPs in ACTN3 and MTHFR with sarcopenia in obese elderly women, after adjusting for age and sex.
The major concern is about the limited study sample size resulting in small and unbalanced subgroups and the modest associations detected in this study. Important clarifications are needed in the Method section to make sure the assumptions of the statistical methods are not violated. The authors should also better justify how the limited number of SNPs in their genetic analyses was selected.
Below are further comments the authors should consider to improve the manuscript.

Author’s response: Please find our responses below – where we tried to address all comments.

Major
-   1) The total sample size of the study is small and dividing the data in four categories make some groups very small (one group has even less than 10 participants). The authors should provide information on how they verified that all assumptions of ANCOVA or logistic regressions were met before analysis. Have the authors also done some power calculations based on their sample size?

Author’s response: Thank you for these comments. We have now done a Shapiro-Wilk test to test whether the data were normally distributed. In the case of violation of normal distribution assumption, the data were log-transformed and then analysed with an ANCOVA. These are mentioned in the revised version to state if the assumptions are met for the analyses, and it reads as following:

“A Shapiro-Wilk test was performed to test whether the data were normally distributed. Variables that were not normally distributed (biceps brachii thickness, MVCEF and MVCKE/VLACSA) were log-transformed before further statistical analysis.”

It should be noted that even though one group is rather small the ANCOVA does consider all individuals to assess an effect of sarcopenia and/or obesity. So, an effect of obesity is essentially a test of all obese vs non-obese people, irrespective of sarcopenia. Likewise, a test of sarcopenia is a test of sarcopenia vs non-sarcopenia, irrespective of obese status. An interaction indicates that the effect of obesity differs in sarcopenic and non-sarcopenic people. This thus enhances the power of detecting statistical effects of obesity and sarcopenia. Nevertheless, we have also done a power analysis, on MVCEF, which revealed that we needed at least 21 people in a group to detect a 23.4 % difference between groups with an α <0.05 and a β>0.80.

We have given this information now in the Methods/limitation section.

-    2) The authors did not mention or perform any correction for multiple testing (Bonferroni or FDR). This should be discussed/justified.

Authors response:  We appreciate the comment received by the reviewer regarding the use of multiple testing. The reason that we have not used multiple testing correction in the present study is due to the presence of prior literature that has already established associations of the selected 24 SNPs with muscle and performance related phenotypes. Thus, a prior hypothesis was generated for each, independent of the others. Had the SNPs never been tested for similar phenotypes, we would have done multiple testing as suggested, because that would have been more speculative and more susceptible to type 1 statistical error, as would randomly testing other variables (genetic or otherwise) not previously associated with the phenotypes of interest.

-    3) The authors need to provide more details on how the 24 SNPs were selected (studies listed in Suppl Table 1). What were the criteria chosen by the authors to select these publications? Why the largest GWAS of handgrip, sarcopenia, lean body mass, and muscle weakness were not included?

Author’s response:  We have added the information requested by the reviewer in the revised version that reads:

“Twenty-four SNPs were selected for the present study (Supplementary Table 1) based on a number of reasons, including the presence of extant literature associating those SNPs with similar phenotypes (even if in different populations), and the likelihood of affecting relevant muscle phenotypes via transcriptional differences as reported in previous studies. Some SNPs were also included due to conflicting results regarding associations with relevant phenotypes.”

We partially agree with the reviewer regarding other SNPs that could have been included such as those identified by previous/recent GWAS for related phenotypes. We did include several such as the TRHR variant, that was associated with lean body mass via GWAS (https://www.sciencedirect.com/science/article/pii/S0002929709000664). However, others were not included including some from more recent large GWAS (e.g. https://www.nature.com/articles/ncomms16015, https://onlinelibrary.wiley.com/doi/full/10.1111/acel.12468), partly because those studies were published after we began our project and purchased relevant assays. We certainly acknowledge that testing other SNPs would be valuable, and have added a sentence to emphasise that our investigation was limited to the 24 studied SNPs, in the paragraph about limitations that preceded the conclusion:

“Finally, this investigation is limited to the 24 studied SNPs, and in fact it is likely that many other SNPs and other types of genetic variants, which we did not investigate, are also associated with the phenotypes we assessed.”

-    4) Limitation section of the Discussion: The authors should discuss more how the small sample sizes and the unbalanced distribution of participants across groups may have affected their results. They should also mention that the study was conducted only in Caucasian participants and thus the results may not generalize to other ancestries or ethnicities.

Authors response: We have included the required information in the revised version in the limitation part. It now reads as:

“The current study has some other limitations. Firstly, participants are distributed unevenly between groups meaning the sarcopenic non-obese group possess few participants. This uneven group distribution could be overcome with a larger population group. Based on a power analysis for MVCEF, we would need at least 21 people in a group to detect a 23.4% difference between groups with an α <0.05 and a β>0.80. Our overall sample size would exceed n = 900. Secondly, one needs to be cautious to extend our observations to men, as the current study is focused on elderly women, although it is likely that the action of most gene variants will not differ between men and women. Our decision to limit the investigation to a single sex can be justified by the earlier onset of sarcopenia-related problems with daily life in women than men [93]. The same argument applies to the fact that we only studied Caucasians, and findings may not be generalized to people with different geographic ancestry. We contend that this is a limitation of extrapolation of results, rather than a real limitation of method, as the genetic differences between people within a geographic ancestry group are larger than the typical genetic differences between geographic ancestry groups [94]. The current study uses BIA for the estimation of skeletal muscle mass, a technique that is reported to overestimate skeletal muscle mass by 0.6 kg in the elderly [95] Despite this shortcoming we used BIA as it is a cheap, accessible and radiation-free method that has been validated with DXA [96], and adopted on numerous occasions for large population studies of elderly muscle mass [97, 98]. Finally, this investigation is limited to the 24 studied SNPs, and in fact it is likely that many other SNPs and other types of genetic variants, which we did not investigate, are also associated with the phenotypes we assessed.

-    5) Abstract and Conclusion section: “Three gene variants linked to affect muscle function, homocysteine metabolism or DNA methylation respectively”. Are the MTFHR SNPs methylation QTLs or is it the gene itself that has been reported by EWAS?

Author’s response: A link between rs1801131 and homocysteine level has been reported previously. None of the variants we observed significant in the current study has been explained as methylation QTLs (so far), however, the previous literature has shown that MTHFR gene has strong epigenetic control and thus the present authors expect those variants may affect the methylation process and thus can regulate gene expression of some genes and thus influence the sarcopenia status.

-    6) The authors used Pearson correlations to identify relevant covariates for BF and SMI. They only considered two (age and height). Height is already included in the definition of SMI. They could have used more sophisticated methods that can account for the joint effect of multiple covariates at the same time. The authors did not provide the results of the Pearson correlation analysis for sarcopenia in the manuscript.

Author’s response: Thank you for pointing out that height is indeed already incorporated in SMI, and therefore we agree it is better to not include it as a covariate, where we are at risk of circular reasoning. We are, however, not sure what sort of ‘more sophisticated methods’ the reviewer refers to, and we think the current analysis does consider the potential impact of biasing factors (co-variates) adequately.

-    7) Line 256: “A possible explanation is that the previous study reported used a different definition of sarcopenia”. Could the authors clarify how the two definitions of sarcopenia differ and why they changed their definition of sarcopenia?

Author’s response: We indeed used different definitions of sarcopenia in the previous and current manuscript. It should be noted that the aim of the previously published paper was to use several previously established sarcopenia definitions and assess how SNPs are associated with those definitions. Here we did not use several previously suggested definitions of sarcopenia but used hand grip strength as suggested by EWGSOP combined with a cut-off for the SMI to define whether a person was sarcopenic or not in our population. In other words, we developed a definition specific to our population of older women.

-    8) More information is needed in the abstract about the statistical analyses performed and the selection of SNPs.

Author’s response: We have added the required information in the abstract in the revised version that reads as following.

“Genotyping was performed for 24 single nucleotide polymorphisms (SNPs), selected based on their previous associations with muscle-related phenotypes.”

“A two-way analysis of covariance was used to assess the main effects of sarcopenia and obesity on muscle-related phenotypes and binary logistic regression was performed for each SNP to investigate associations with sarcopenia in obesity.”

Minor
-    9) Abstract and Results section: “There were no significant obesity * sarcopenic status interaction for any of the parameters” please rephrase/clarify what are these parameters

Author’s response: We have added “investigated muscle-related phenotypic parameters” in the revised version to clarify.

-    10) Please replace the following terminology in the paper: “linked” and “interlinked”

Author’s response: We have used both of those terminologies with caution throughout the manuscript.

-    11) In the description of the genetic analyses in the Method section, the authors did not mention that the analyses were performed only in obese women. Please add this information.

Authors response: Thank you for the comment. We have added the missing information in the revised version that reads as follows-

“Binary logistic regression was performed for each individual SNP using the covariates to investigate their association with sarcopenia in the obese elderly.”

-    12) Suggestion to define sarcopenia in the abstract

Authors response: The addition of defining sarcopenia in the abstract may lead to define obesity as well in the abstract. Therefore, we believe that defining sarcopenia may be somewhat not necessary in the abstract, but we define both sarcopenia and obesity with clarity in the method – that is very necessary in this manuscript as the reviewer pointed out.

-    13) Suggestion to use “describe” instead of “identify” for the first sub-aim in the abstract, as was done at the end of the introduction

Authors response: We have used those words as per the reviewer suggestion in the revised version.

-    14) Lines 175-176: “A two-way Analysis of covariance (ANCOVA) was used to assess the main effects of sarcopenia and obesity “. The outcomes are missing (effect of X on Y). Please add this information.

Authors response: Added in the revised version and it reads as –

“A two-way Analysis of covariance (ANCOVA) was used to assess the main effects of sarcopenia and obesity on muscle-related phenotypic outcome measures, where a sarcopenia * obesity interaction would indicate that the effects of sarcopenia differed between the obese and non-obese, or that the effect of obesity was different in sarcopenic and non-sarcopenic women.”

-    15) Line 201 “indicating that all observed effects of sarcopenia are the same” please rephrase

Authors response: Thank you for the comment. The original text has been rephrased to the following:

16) Line 248: “Our findings on muscle function and muscle quality suggest that sarcopenia and obesity act synergistically” please use another term as no interaction was significant.

Authors response: Based on the re-analyses, we observed there was no interaction of sarcopenia and obesity for any phenotype. Therefore, we have removed the word synergistically in the revised version.

-    17) Table 2: please clarify what the numbers in each cell (column 1-4) represent (mean (95% CI)), and maybe also consider revising the title.

Reviewer 2 Report

Interesting study and well written manuscript.

Study design good, results may well facilitate early detection of sarcopenic obesity which would be hugely beneficial.

Limitations in methods well discussed, and my major concern was one group had a very low n number but this is addressed and discussed. The three gene varients they discuss are different from a very similar previous study, and whilst this is dicussed I feel it should be expanded upon. Could it also be due to detection methods, difference in cohorts, reliability of datasets? Too broad a definition of sarcopenia and therefore too little overlap betweem different measurements? Not necessarily discussed futher in the text but a response written for reviewers. Aside from this, the results are novel, the data presented well and well written manuscript. 

Author Response

Reviewer 2:

  • Interesting study and well written manuscript.

Study design good, results may well facilitate early detection of sarcopenic obesity which would be hugely beneficial.

Limitations in methods well discussed, and my major concern was one group had a very low n number but this is addressed and discussed. The three gene varients they discuss are different from a very similar previous study, and whilst this is dicussed I feel it should be expanded upon. Could it also be due to detection methods, difference in cohorts, reliability of datasets? Too broad a definition of sarcopenia and therefore too little overlap betweem different measurements? Not necessarily discussed futher in the text but a response written for reviewers. Aside from this, the results are novel, the data presented well and well written manuscript. 

Author’s response: Thank you for liking our manuscript based on novelty of the findings and how it is interpreted and written.

As both reviewers commented regarding sample size and one group having fewer participants, we have responded to the other reviewer regarding this and refer this reviewer to those comments above, that we hope to address the issue satisfactorily.

Similarly, we have also addressed the use of different sarcopenia definitions between this manuscript and the previously published paper (https://www.nature.com/articles/s41598-020-59722-9) in our responses to Reviewer 1.
